# BIDIRECTIONAL LEARNING FOR THE VISUAL REPRESENTATION IN RADIOLOGY REPORT GENERATION WITH FROZEN LLMS

## ABSTRACT

Radiology report generation (R2Gen) has recently leveraged large language models (LLMs), achieving improved results. However, the generated reports still fall short in both language accuracy and clinical relevance. A key challenge is learning a visual representation of radiology images that an LLM can effectively interpret. To address this, we propose that for a visual representation to be interpretable by an LLM, it shall also be generatable by the LLM. Building on this idea, we introduce a novel bidirectional learning framework for R2Gen, integrating both vision-to-text and text-to-vision information to enhance visual representation learning. First, we require that the visual representation aid the LLM in generating reports that closely match the ground truth. Second, we require that the visual representation be maximally generated by the LLM when provided with the ground truth report. To enable the frozen LLM to perform text-to-vision generation, we jointly train a new text encoder for reports. Additionally, through an image reconstruction task, we encourage the visual representation to capture the core features of input radiology images. This bidirectional learning framework is realized using a frozen LLM and incurs no extra computational cost at the inference stage. Experimental results demonstrate better alignment between the learned visual representation and the LLM's word embedding space, along with state-of-the-art performance in both language accuracy and clinical efficacy. Our code will be publicly released.

## 1 INTRODUCTION

Automated radiology report generation (R2Gen) has emerged as a promising solution to alleviate the heavy workload of radiologists while ensuring the consistency and accuracy of medical reports. Significant progress has been made in this area, with numerous methods proposed in the literature Chen et al. (2020; 2022); Liu et al. (2021a); Yang et al. (2022). The recent development of large language models (LLMs) has further advanced this area, enabling the generation of more accurate reports thanks to their exceptional natural language generation (NLG) capabilities Wang et al. (2023b); Yang et al. (2023); Lee et al. (2023). However, despite their state-of-the-art performance, current LLM-based R2Gen methods continue to face key challenges in achieving the level of linguistic accuracy and clinical efficacy necessary for real-world medical diagnosis.

Among the various challenges, a key issue in LLM-based R2Gen methods is improving the effectiveness of the visual representation that is fed into the LLM to generate reports. Since this representation provides all the information that the LLM can have about the input radiology image (e.g., chest X-ray images), its effectiveness is crucial. Compared with existing approaches that incorporate additional or external information (e.g., lesion-aware augmentations Hou et al. (2024), structured clinical information Dalla Serra et al. (2022), or knowledge bases Huang et al. (2023); Ranjit et al. (2023); Jin et al. (2024a); Bu et al. (2024)) to address this issue, this work takes a different approach that fully exploits the available image-report training pairs to enhance visual representation learning, without introducing any supplementary information.

In this work, we focus on enhancing the compatibility of the visual representation with the LLM, ensuring it is more easily "understood" by the LLM. This challenge, commonly noted in the literature, arises from the well-known modality gap between vision (captured by the visual representation) and

text (the modality on which the LLM is trained). A typical remedy to this case is to use a visual mapper to project visual features onto the space where the LLM's word embedding space resides. However, this remedy alone often falls short in fully bridging the gap between visual and textual modalities, limiting the LLM's ability to fully understand the visual input. To further address this gap, CLIP-based contrastive learning is a natural extension to align visual embeddings with text embeddings more closely. Differently, in this work, we propose a more radical approach.

Our work is inspired by Richard Feynman's famous quote: "*What I cannot create, I do not understand.*" We propose that for a visual representation to be truly "understood" by an LLM, it shall be generatable by the LLM. This requirement is used to achieve a stronger compatibility between the visual representation and the LLM's word embedding space. Furthermore, the generation of the visual representation by the LLM should not be done in a random or arbitrary manner. Instead, considering the correspondence between each image and its ground truth report in the training data, we require that the visual representation be generated when the report is used as input to the LLM.

Building on this idea, we propose a novel bidirectional learning framework to learn visual representations for radiology report generation. It is fully built around a frozen LLM, ensuring computational efficiency and avoiding performance degradation from improper fine-tuning. This framework involves both vision-to-text and text-to-vision tasks. For the vision-to-text task, we require the visual representation to enable the LLM to generate a report that closely matches the ground truth. In an innovative twist, the text-to-vision task requires the visual representation to be maximally generatable by the LLM when provided with the ground truth report. Considering that the LLM, having been trained primarily on textual data, may not effectively handle a text-to-vision task with its built-in word embedding, we jointly learn a new text encoder between the ground truth report and the frozen LLM. Lastly, to enhance the visual representation further, we require it to support the reconstruction of the input radiology image, enabling it to capture the core characteristics of the underlying distribution of these radiology images.

To further understand the core principles of this visual representation learning, we analyze it through the lens of model regularization. From this perspective, the text-to-vision task functions as a regularization constraint, preventing the visual encoder from becoming overly complex and enhancing its generalization capabilities. Once trained, the framework can generate a report when a radiology image is provided. Importantly, at the inference stage, only the vision-to-text branch is needed, operating in the same way as current LLM-based R2Gen methods. This ensures that our method does not introduce any additional computational overhead when deployed.

Extensive experiments on the IU-Xray and MIMIC-CXR datasets demonstrate that the proposed framework consistently outperforms existing methods, achieving the state-of-the-art results in both language accuracy and clinical efficacy. Additionally, ablation studies validate the contribution of the proposed bidirectional learning process, confirming the improved compatibility between the learned visual representation and the LLM's word embedding space.

## 2 RELATED WORK

**Radiology Report Generation.** Radiology report generation (R2Gen) has traditionally focused on vision-to-text architectures, where a vision encoder extracts features from CXR scans and a text decoder produces the corresponding report. Early work in this domain used CNNs and LSTMs Jing et al. (2017); Wang et al. (2018); Xue et al. (2018), followed by transformer-based models that significantly improved report quality Chen et al. (2020; 2022); Nicolson et al. (2023); Wang et al. (2023b); Lee et al. (2023). Despite these advancements, achieving language accuracy and clinical efficacy remains a persistent challenge.

To improve report generation quality, several approaches have been explored. Some methods focus on detecting and describing key anatomical regions Tanida et al. (2023); Dalla Serra et al. (2023), while others emphasize feature alignment between visual and textual modalities Wang et al. (2022; 2023a); Li et al. (2023). Additionally, external knowledge sources, such as lesion-aware augmentations Hou et al. (2024), structured clinical information Dalla Serra et al. (2022), symptom graphs, or knowledge distiller Liu et al. (2021b); Huang et al. (2023); Ranjit et al. (2023); Jin et al. (2024a); Bu et al. (2024), have shown promise. However, constructing specialized knowledge bases for such integration remains resource-intensive and demands domain expertise.

Most of aforementioned methods follow a unidirectional vision-to-text approach, focusing on visual feature extraction while not adequately utilizing the semantic richness embedded in radiology reports. In contrast, our approach fully leverages report semantics without introducing external knowledge or supplementary data.

The recent work MedM2G Zhan et al. (2024) aims to unify multiple cross-modality medical generation tasks (e.g., among text, X-ray, CT, and MRI) into a single framework Rombach et al. (2022). Our work differs from it in several ways. First, MedM2G focuses on unifying cross-modality generation tasks across different medical imaging modalities, while we concentrate primarily on improving radiology report generation. Second, MedM2G uses the integration of multiple diffusion models, whereas our framework is based on the LLM-based R2Gen setting. Third, MedM2G does not explicitly explore bidirectional learning for aligning images and reports. At last, as will be shown in the experiment, our method achieves better performance on radiology report generation benchmarks.

**LLM-based R2Gen Models.** Recent large language models (LLMs) have demonstrated impressive abilities in text generation across various fields, offering new potential for improving R2Gen systems. For instance, RAG Ranjit et al. (2023) treats R2Gen as a retrieval task, utilizing the GPT-3.5-turbo and GPT-4 to generate reports based on a retrieval corpus. R2GenGPT Wang et al. (2023b) uses a frozen LLM to generate reports and map visual features to the LLM built-in embedding space. It improves language fluency, yet clinical accuracy remains a key challenge. Other works, like MedXChat Yang et al. (2023) and LLM-CXR Lee et al. (2023), fine-tune LLMs to consolidate multiple tasks, such as image-to-report generation and visual question answering (VQA) into one unified framework. While MedXChat incorporates instruction tuning and fine-tunes Stable Diffusion Rombach et al. (2022) for CXR report generation and image synthesis, LLM-CXR focuses on image understanding and generation by tokenizing chest X-ray images through VQ-GAN Esser et al. (2021) and using instruction-finetuning to perform tasks such as CXR generation and CXR-related VQA. Both methods leverage the instruction-following capabilities of LLMs to connect text and image modalities in medical applications.

Our approach differs from MedXChat and LLM-CXR in several ways: (1) Instead of fine-tuning LLMs, we keep the LLM frozen and focus on improving visual representations without modifying the LLM; (2) As will be shown, our method achieves better performance than MedXChat in radiology report generation. As for LLM-CXR, it concentrates on image synthesis and VQA and does not provide strictly comparable results for radiology report generation; (3) Different from their methods where vision-to-text and text-to-vision tasks operate independently, our bidirectional learning approach ensures that text-to-vision generation enhances the visual encoder's effectiveness, leading to improved report quality.

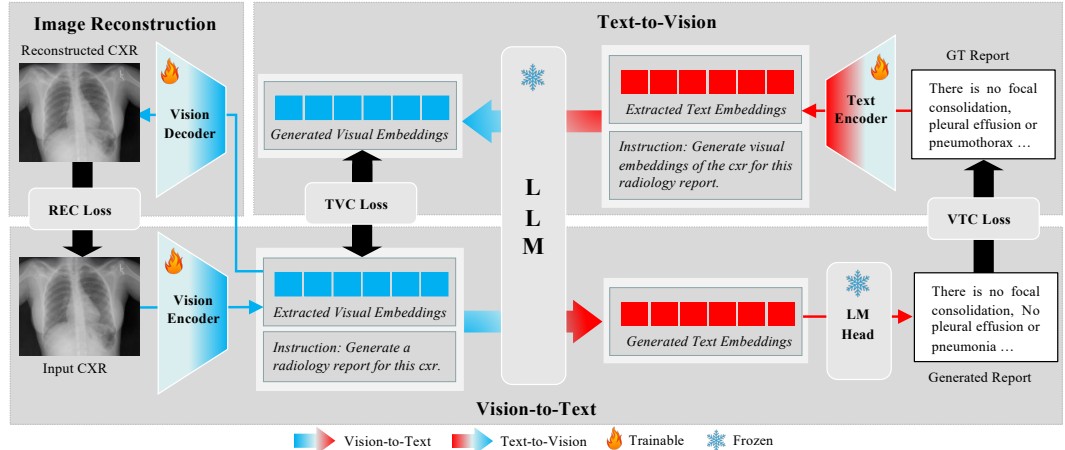

Figure 1: Overview of the proposed bidirectional learning framework for radiology report generation. It consists of three components: 1) the vision-to-text branch; 2) the text-to-vision branch; 3) the image reconstruction component. To keep the diagram concise, certain detailed components, such as the visual and textual mappers, have been omitted. Details are provided in Section 3.

## 3 METHODOLOGY

**Framework Overview.** As illustrated in Figure 1, the proposed framework consists of three components, corresponding to the vision-to-text task (at the bottom part of the figure), the text-to-vision task (at the top-right part), and the image reconstruction task (at the top-left corner). At the algorithm level, the central issue is to optimise the network parameters of the vision encoder. It is conducted by minimising a combination of three loss terms. They are presented in order as follows.

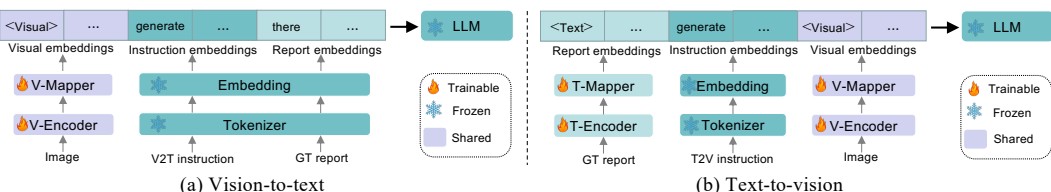

(a) Vision-to-text                (b) Text-to-vision

Figure 2: The format of the input to the LLM for the vision-to-text (a) and text-to-vision (b) tasks. In both cases, the embedding layer and tokenizer of the frozen LLM (i.e., the parts in dark green) are shared between the branches, while trainable encoders and mappers are specific to each modality. The vision encoder and visual mapper, along with the extracted vision embeddings, are shared between the two branches (indicated in purple). Note that "V-Mapper" and "T-Mapper" are short for visual and textual mappers, respectively.

**Vision-to-Text Branch.** As shown in Figure 2(a), given an input image $I$, the vision encoder produces its visual features $\mathbf{Z}_v = g_v(I; \boldsymbol{\theta}_v)$, where $\mathbf{Z} \in \mathbb{R}^{N_p \times d_v}$, with $N_p$ being the number of patches and $d_v$ the dimensionality of the visual features. Then a visual mapper is used to project the visual features onto the space where the LLM's word embedding space resides. That is, $\mathbf{E}_v = m_v(\mathbf{Z}_v)$, where $\mathbf{E}_v \in \mathbb{R}^{N_p \times d_L}$, $d_L$ denotes the dimensionality of the LLM's word embedding space, and $m_v(\cdot)$ represents the trainable visual mapper. The visual embeddings obtained in $\mathbf{E}_v$ are fed into the frozen LLM to generate a report, instructed by a prompt denoted by $\mathbf{S}_{v2t}$, as illustrated in Figure 3(a). The generated report is compared with the ground-truth report through a vision-text consistency (VTC) loss, defined as the auto-regressive negative log-likelihood of generating the correct report tokens. It is expressed as

$$\mathcal{L}_{\text{VTC}} = -\sum_{k=1}^{l} \log P(t_k \mid \mathbf{E}_v, \mathbf{S}_{v2t}, T_{<k}), \tag{1}$$

where $l$ is the length of a report and $T_{<k}$ represents the ground truth tokens preceding the token $t_k$ to be generated. Through error back-propagation, this loss encourages the vision encoder to produce visual features that best support the LLM to generate high-quality reports.

**Text-to-Vision Branch.** The text-to-vision branch works with the vision-to-text branch to form the proposed bidirectional learning process. It further tightens the connection between the extracted visual embeddings $\mathbf{E}_v$ and the ground truth reports. This process is powered by the LLM, which plays a central role in extracting the rich semantic information in the reports and converting it into visual representations. Nevertheless, two subtle issues needs to be addressed beforehand.

One issue is that radiology reports often omit whether the CXR image is frontal or lateral, leading to potential mismatches when generating visual embeddings. To avoid this, we train a classifier using only CXR images to identify the view (frontal or lateral) and append this information to the report. This ensures that the LLM generates visual embeddings aligned with the correct view. The classifier is trained strictly on the training set images and serves as a preprocessing step, without influencing the model's training or testing.

The other issue, which is more important, is that we need to jointly learn a new text encoder between the ground truth report and the frozen LLM, instead of directly using the LLM's built-in word embedding layer. This is due to several considerations: 1) the LLM, having been trained primarily on textual data, may not effectively handle a text-to-vision task with its built-in word embedding layer; 2) the text encoder can be optimized to best support the LLM to generate the targeted visual embeddings; and 3) this design ensures the LLM remains frozen, preserving its original capabilities.

As shown in Figure 2(b), the ground truth report $T$, now containing the view information, is fed into a text encoder to extract textual features $\mathbf{Z}_t = g_t(T; \boldsymbol{\theta}_t)$, where $\mathbf{Z}_t \in \mathbb{R}^{l \times d_t}$ with $l$ being the length of a report and $d_t$ the dimensionality of the textual features. As in the vision-to-text branch, a trainable textual mapper $m_t(\cdot)$ is used to project the textual features onto the space where the LLM's word embeding space resides, and this produces the text embeddings $\mathbf{E}_t = m_t(\mathbf{Z}_t)$. The embeddings are then fed into the frozen LLM to generate the visual embeddings denoted by $\mathbf{E}_g \in \mathbb{R}^{N_p \times d_L}$. Recall that $N_p$ is the number of image patches determined by the vision encoder $g_v(\cdot)$. Note that the LLM is instructed by a text-to-vision prompt, in which the instruction is $\mathbf{S}_{t2v}$ and the visual embeddings (extracted by the vision encoder and projected by the visual mapper) in $\mathbf{E}_v$ are provided as the target value, as indicated by the purple segment in Figure 2(b). This requires the learning process to optimise the new text encoder to enable the LLM to generate target visual embeddings. Formally, this is conducted via the text-vision consistency (TVC) loss as follows.

$$\mathcal{L}_{\text{TVC}} = \frac{1}{N_p} \sum_{i=1}^{N_p} \|\mathbf{E}_{v,i} - \mathbf{E}_{g,i}\|^2. \tag{2}$$

where the subscript "$i$" indicates the $i$th vector in the embedding matrix $\mathbf{E}$.

**Image Reconstruction.** This component requires the visual embeddings, $\mathbf{E}_v$, to support the reconstruction of the input radiology image. We firstly use a trainable mapper $m_v'(\cdot)$ to project $\mathbf{E}_v$ from the LLM's word embedding space back to the output space of the vision encoder, that is, $\hat{\mathbf{Z}}_v = m_v'(\mathbf{E}_v)$, where $\hat{\mathbf{Z}}_v \in \mathbb{R}^{N_p \times d_v}$. After that, $\hat{\mathbf{Z}}_v$ is fed into the vision decoder $g_{vd}$ to reconstruct the image as $\hat{I} = g_{vd}(\hat{\mathbf{Z}}_v; \boldsymbol{\theta}_{vd})$, where $\boldsymbol{\theta}_{vd}$ represents the trainable network parameters of $g_{vd}$. The reconstructed error is measured by the discrepancies between corresponding pixels as $\mathcal{L}_{\text{REC}} = \frac{1}{CHW} \sum_{c=1}^{C} \sum_{i=1}^{H} \sum_{j=1}^{W} \|I_{c,i,j} - \hat{I}_{c,i,j}\|^2$, where $H$, $W$, and $C$ are the height, width, and channel numbers of the input CXR image, and $I_{c,i,j}$ and $\hat{I}_{c,i,j}$ represent the pixel values at channel $c$ and position $(i,j)$ of the original and reconstructed images. Minimizing the reconstruction loss encourages the vision encoder to extract the visual representation that can characterise the essential features of the training images.

**Loss Function and Interpretation.** For a given training sample in the form of an image-report pair $(I, T)$, its total loss function is

$$\mathcal{L}_{\text{TOTAL}}(I, T) = \mathcal{L}_{\text{VTC}} + \lambda_1 \mathcal{L}_{\text{TVC}} + \lambda_2 \mathcal{L}_{\text{REC}}, \tag{3}$$

where $\lambda_1$ and $\lambda_2$ are hyperparameters that balance the contributions of each loss term.

To interpret this loss, we specially highlight the visual embedding $\mathbf{E}_v$ and its dependence on the network parameter $\boldsymbol{\theta}_v$ of the vision encoder, and we temporarily omitting the parameters of the visual and textual mappers. The total loss is then rewritten as

$$\mathcal{L}_{\text{TOTAL}}(I, T) = \underbrace{\mathcal{L}_{\text{VTC}}(\mathbf{E}_v(\boldsymbol{\theta}_v))}_{\text{Error term}} + \underbrace{\lambda_1 \mathcal{L}_{\text{TVC}}(\mathbf{E}_v(\boldsymbol{\theta}_v), \boldsymbol{\theta}_t) + \lambda_2 \mathcal{L}_{\text{REC}}(\mathbf{E}_v(\boldsymbol{\theta}_v), \boldsymbol{\theta}_{vd})}_{\text{Regularization terms}}. \tag{4}$$

As indicated, we interpret the first term on the right hand side as an "error term" because it corresponds to the training error measured by the negative log-likelihood of generating the correct report tokens, which is essentially a cross-entropy classification loss. We interpret the following two terms as "regularization terms" by considering that they impose penalties on the objective function (i.e., the total loss) to constrain the space where $\mathbf{E}_v$ (or more fundamentally, the vision encoder's parameter $\boldsymbol{\theta}_v$) can reside. Now we focus on the term $\mathcal{L}_{\text{TVC}}$, which is the key contribution of this work, to further interpret its regularization effect by rewriting Eq.(2) in a compact form as

$$\mathcal{L}_{\text{TVC}}(\mathbf{E}_v(\boldsymbol{\theta}_v), \boldsymbol{\theta}_t) = \frac{1}{N_p} \|\mathbf{E}_v(\boldsymbol{\theta}_v) - \mathbf{E}_g(\boldsymbol{\theta}_t)\|_{\mathcal{F}}^2, \tag{5}$$

where $\mathcal{F}$ denotes the matrix Frobenius norm. Now it is clear that $\mathcal{L}_{\text{TVC}}$ essentially imposes a prior on $\mathbf{E}_v(\boldsymbol{\theta}_v)$, which is $\mathbf{E}_g(\boldsymbol{\theta}_t)$, and measures its deviation from this prior. More interestingly, this prior is not a predefined, static one but optimised to adaptively vary with the ground truth report. Geometrically, $\mathbf{E}_v(\boldsymbol{\theta}_v)$ is restricted within a high-dimensional sphere centered at $\mathbf{E}_g(\boldsymbol{\theta}_t)$ while trying to move towards the low-value regions of the error term $\mathcal{L}_{\text{VTC}}$. During optimising the total loss, $\mathcal{L}_{\text{TVC}}$ negotiates with $\mathcal{L}_{\text{TVC}}$ to find the optimal $\boldsymbol{\theta}_v$ (and $\boldsymbol{\theta}_t$).

In addition to the above regularization perspective, we could also interpret $\mathcal{L}_{\text{TVC}}$ in Eq.(5) as another type of alignment between an image, represented by $\mathbf{E}_v(\boldsymbol{\theta}_v)$, and its ground truth report, whose semantic information is conveyed by $\mathbf{E}_g(\boldsymbol{\theta}_t)$. Such alignment takes place in the space where the LLM's word embedding space resides and the alignment degree is measured by the distance between them. In sum, the introduction of $\mathcal{L}_{\text{VTC}}$ and $\mathcal{L}_{\text{REC}}$, either interpreted from the perspective of regularization or alignment, further constrains the feasible domain of the vision encoder's parameter $\boldsymbol{\theta}_v$. According to regularization theory Haykin (2007) (Chapter 7), this helps to reduce the model complexity of the vision encoder and improve its generalization capability.

## 4 Experiments

### 4.1 Datasets and Settings

**IU-Xray.** Indiana University Chest X-ray Collection (IU-Xray) Demner-Fushman et al. (2016) contains 3,955 de-identified radiology reports, each of which is associated with frontal and/or lateral chest X-ray images, and 7,470 chest X-ray images in total. Each report is comprised of several sections: Impression, Findings, Indication, etc. In this work, we adopt the same data set partitioning as in the literature Chen et al. (2020) for a fair comparison, with a training/validation/test split set by 7:1:2 of the entire dataset. All evaluations are done on the test set.

**MIMIC-CXR.** This dataset includes 377,110 chest X-ray images from 65,379 patients, each accompanied by a radiology report. We adhere to the training/validation/test split in the literature Chen et al. (2020) to facilitate comparison with related methods. This split results in 270,790 images for training, 2,130 for validation, and 3,858 for testing, all paired with their respective reports. Keeping the aspect ratio, all images are resized with the shorter edge to be 256 pixels.

**Evaluation Metrics.** We utilize standard natural language generation (NLG) metrics for assessment, including BLEU scores Papineni et al. (2002), ROUGE-L Chin-Yew (2004), and METEOR Banerjee & Lavie (2005). Specifically, to evaluate the model, we use the BLEU-4 score on the validation set to select the optimal model checkpoint. In addition to the NLG metrics, we employ several clinical efficacy (CE) metrics to assess the model's ability to generate clinically accurate reports. These include the RadCliQ metric Yu et al. (2023), which combines multiple individual metrics to align with radiologist evaluations. We also utilize the Bert Score Zhang et al. (2019), which measures the semantic similarity between generated reports and ground-truth reports by comparing contextualized embeddings, and the RadGraph F1 score Jain et al. (2021), which evaluates the model's ability to correctly extract and describe clinical entities and their relationships. These CE metrics provide a comprehensive assessment of clinical accuracy.

**Implementation Details.** The Llama2-7B model [1] is utilised as text decoder, and the text encoder of the base version of the BLIP [2] is used as the new text encoder. The large version of the MAE [3] forms the visual part, i.e., visual encoder and decoder. Between the LLM and the visual text encoders, there is a fully connected layer that can be used to project visual text features to match the dimensions of the LLM's embedding space. Training is conducted on four NVIDIA A100 80GB GPUs for 5 epochs on MIMIC-CXR and 2 epochs on IU-Xray, with a mini-batch size of 4, a learning rate of 1e-4, AdamW optimizer Loshchilov & Hutter (2018), and a cosine annealing scheduler. Testing involves beam search with a size of 3. Images are randomly cropped to 224 × 224 during training and inference. Both $\lambda_1$ and $\lambda_2$ in equation 3 are set as 1 without fine-tuning.

### 4.2 Results and Discussion

Table 1 compares the proposed method and relevant state-of-the-art ones on the IU-Xray and MIMIC-CXR datasets. This comparison includes image captioning methods such as Show-Tell Vinyals et al. (2015), AdaAtt Xu et al. (2015), and M2Transformer Cornia et al. (2020), medical report generation methods like R2Gen Chen et al. (2020), R2GenCMN Chen et al. (2022), PP-KED Liu et al. (2021a), GSK Yang et al. (2022), MSAT Wang et al. (2022), METransformer Wang et al. (2023a), CvT2DistilGPT2 Nicolson et al. (2023), KGEER Dalla Serra et al. (2022), D$^2$-Net Jin

---

[1]https://huggingface.co/meta-llama/Llama-2-7b-chat-hf

[2]https://huggingface.co/Salesforce/blip-image-captioning-base

[3]https://huggingface.co/facebook/vit-mae-large

Table 1: Comparison on IU-Xray (upper part) and MIMIC-CXR datasets (lower part). † indicates the results are quoted from their respective papers. Those without † are obtained by re-running the publicly released codebase Li et al. (2021) using the same training-test partition as our method. The highest and second highest performance are highlighted by bolding and underlining respectively.

| Dataset | Methods | BLEU-1 | BLEU-2 | BLEU-3 | BLEU-4 | ROUGE | METEOR |
|---|---|---|---|---|---|---|---|
| **IU-Xray** | Show-Tell | 0.243 | 0.130 | 0.108 | 0.078 | 0.307 | 0.157 |
| | AdaAtt | 0.284 | 0.207 | 0.150 | 0.126 | 0.311 | 0.165 |
| | M2Transformer | 0.402 | 0.284 | 0.168 | 0.143 | 0.328 | 0.170 |
| | R2Gen† | 0.470 | 0.304 | 0.219 | 0.165 | 0.371 | 0.187 |
| | R2GenCMN† | 0.475 | 0.309 | 0.222 | 0.170 | 0.375 | 0.191 |
| | MSAT | 0.481 | 0.316 | 0.226 | 0.171 | 0.372 | 0.190 |
| | CvT2DistilGPT2† | 0.473 | 0.304 | 0.224 | 0.175 | 0.376 | 0.200 |
| | R2GenGPT† | 0.488 | 0.316 | 0.228 | 0.173 | 0.377 | 0.211 |
| | Ours | **0.512** | **0.341** | **0.249** | **0.186** | **0.392** | **0.221** |
| | *Results below are not strictly comparable due to different data partition. For reference only.* | | | | | | |
| | PPKED† | 0.483 | 0.315 | 0.224 | 0.168 | 0.376 | 0.187 |
| | METransformer† | 0.483 | 0.322 | 0.228 | 0.172 | 0.380 | 0.192 |
| | D²-Net† | 0.492 | 0.327 | 0.231 | 0.171 | 0.378 | 0.204 |
| | EKAGen† | 0.526 | 0.361 | 0.267 | 0.203 | 0.404 | 0.214 |
| | MedM2G† | 0.533 | 0.369 | 0.278 | 0.212 | 0.416 | - |
| **MIMIC-CXR** | Show-Tell | 0.308 | 0.190 | 0.125 | 0.088 | 0.256 | 0.122 |
| | AdaAtt | 0.314 | 0.198 | 0.132 | 0.094 | 0.267 | 0.128 |
| | M2Transformer | 0.332 | 0.210 | 0.142 | 0.101 | 0.264 | 0.134 |
| | R2Gen† | 0.353 | 0.218 | 0.145 | 0.103 | 0.277 | 0.142 |
| | R2GenCMN† | 0.353 | 0.218 | 0.148 | 0.106 | 0.278 | 0.142 |
| | PPKED† | 0.360 | 0.224 | 0.149 | 0.106 | 0.284 | 0.149 |
| | GSK† | 0.363 | 0.228 | 0.156 | 0.115 | 0.284 | - |
| | MSAT† | 0.373 | 0.235 | 0.162 | 0.120 | 0.282 | 0.143 |
| | METransformer† | 0.386 | 0.250 | 0.169 | 0.124 | 0.291 | 0.152 |
| | UniXGen-256† | 0.365 | 0.227 | 0.147 | 0.101 | 0.294 | 0.156 |
| | CvT2DistilGPT2† | 0.393 | 0.248 | 0.171 | 0.127 | - | 0.155 |
| | KGEER† | 0.363 | 0.245 | 0.178 | 0.136 | 0.313 | 0.161 |
| | D²-Net† | 0.365 | 0.230 | 0.153 | 0.107 | 0.278 | 0.136 |
| | EKAGen† | 0.419 | 0.258 | 0.170 | 0.119 | 0.287 | 0.157 |
| | MedM2G† | 0.412 | 0.260 | 0.179 | 0.142 | 0.309 | - |
| | LLM-CXR† | 0.196 | 0.095 | 0.054 | 0.033 | 0.245 | 0.081 |
| | MedXChat† | 0.367 | 0.235 | 0.158 | 0.111 | 0.264 | 0.135 |
| | R2GenGPT† | 0.411 | 0.267 | 0.186 | 0.134 | 0.297 | 0.160 |
| | Ours | **0.427** | **0.285** | **0.202** | **0.144** | **0.314** | **0.171** |

et al. (2024b), EKAGen Bu et al. (2024), and MedM2G Zhan et al. (2024), as well as LLM-based R2Gen methods LLM-CXR Lee et al. (2023), MedXChat Yang et al. (2023), and R2GenGPT Wang et al. (2023b). Since the IU-Xray dataset lacks an official training-test partition, results for some methods (PPKED, METransformer, D²-Net, EKAGen, MedM2G) are not strictly comparable and are provided for reference only. However, all models on MIMIC-CXR use the official partition, ensuring comparability.

**Language Quality Analysis.** As demonstrated in Table 1, our framework consistently outperforms prior methods across all key metrics. On IU-Xray, among those strictly comparable methods in Table 1, our method improves the second-best method by BLEU-4 from 0.175 (CvT2DistilGPT2) to 0.186, ROUGE from 0.377 (R2GenGPT) to 0.392, and METEOR from 0.211 (R2GenGPT) to 0.221. Among those reference methods (gray part in Table 1), EKAGen and MedM2G show better performance than ours. However, the interpretation of the results requires caution. First, as mentioned, IU-Xray does not provide an official training-test partition, rendering the results not strictly comparable. Second, as IU-Xray is a relatively small dataset, varying training-test partitions can significantly affect performance. Notably, these two methods consistently underperform when strictly compared with our model on the larger MIMIC-CXR dataset, as discussed below.

More importantly, on the MIMIC-CXR dataset, our method is the best performer, delivering notable improvements. BLEU-1 increases from 0.419 (EKAGen) to 0.427, BLEU-2 from 0.267 (R2GenGPT) to 0.285, BLEU-3 from 0.186 (R2GenGPT) to 0.202, and BLEU-4 from 0.142 (MedM2G) to 0.144. ROUGE rises from 0.313 (KGEER) to 0.314, and METEOR from 0.161

(KGEER) to 0.171, further demonstrating the model's superior contextual accuracy. Notably, on this larger dataset, the two most recent methods EKAGen and MedM2G perform similarly to, if not worse than, R2GenGPT that employs a neat structure to incorporate LLMs for report generation, demonstrating the advantages of LLMs in this task. Our model further advances R2GenGPT by i) integrating the proposed bidirectional learning to enhance the compatibility of visual representation and LLM's comprehension and ii) image reconstruction for more regularization. This synergy leads to the significant improvements in report generation quality observed in both datasets. Focusing on unifying multiple multi-modal analysis tasks, MedXChat and LLM-CXR do not outperform the models specially designed for R2Gen.

Table 2: Evaluation of Clinic-related Metrics on MIMIC-CXR.

| Methods | RadGraph F1 ($\uparrow$) | Bert Score ($\uparrow$) | RadCliQ ($\downarrow$) |
|---|---|---|---|
| R2Gen | 0.172 | 0.406 | 1.228 |
| R2GenCMN | 0.182 | 0.418 | 1.182 |
| CvT2DistilGPT2 | 0.196 | 0.374 | 1.220 |
| RaDialog-RG$^\dagger$ | - | 0.40 | - |
| R2GenGPT | 0.187 | 0.415 | 1.207 |
| **Ours** | **0.203** | **0.427** | **1.169** |

**Clinical Efficacy Analysis.** Clinical efficacy scores, like RadGraph F1, Bert Scores, and RadCliQ, are only computable on MIMIC-CXR, shown in Table 2. Our framework achieves the best results across varied clinical efficacy metrics when compared to state-of-the-art methods. Specifically, it improves the RadGraph F1 score, which measures clinical entity extraction accuracy, from 0.196 (CvT2DistilGPT2) to 0.203, showcasing the model's superior ability to capture clinical concepts and their relationships. Additionally, Bert Score, which assesses the semantic similarity between generated and ground-truth reports, increases from 0.418 (R2GenCMN) to 0.427, indicating enhanced report fluency and clinical relevance. For RadCliQ, where a lower score reflects better clinical quality, our framework decreases the score from 1.182 (R2GenCMN) to 1.169, further demonstrating its ability to produce more clinically accurate and error-free reports. These improvements highlight the strength of bidirectional visual representation learning and the image reconstruction in refining the model's clinical efficacy.

Table 3: Ablation study of the components in our model ("-" means not applicable).

| Dataset | V2T | T2V | Img Rec | Epoch | BLEU-4 | ROUGE | METEOR | RadGraph F1 | Bert Score | RadCliQ ($\downarrow$) |
|---|---|---|---|---|---|---|---|---|---|---|
| **IU-Xray** | ✓ | | | 8 | 0.176 | 0.380 | 0.213 | - | - | - |
| | ✓(+CLIP) | | | 6 | 0.178 | 0.383 | 0.211 | - | - | - |
| | ✓ | ✓ | | 5 | 0.184 | 0.388 | 0.219 | - | - | - |
| | ✓ | ✓ | ✓ | 2 | **0.186** | **0.392** | **0.221** | - | - | - |
| **MIMIC-CXR** | ✓ | | | 9 | 0.135 | 0.302 | 0.163 | 0.188 | 0.416 | 1.201 |
| | ✓(+CLIP) | | | 7 | 0.135 | 0.307 | 0.162 | 0.188 | 0.418 | 1.197 |
| | ✓ | ✓ | | 7 | 0.141 | **0.315** | 0.167 | 0.196 | 0.425 | 1.178 |
| | ✓ | ✓ | ✓ | 5 | **0.144** | 0.314 | **0.171** | **0.203** | **0.427** | **1.169** |

**Ablation Study.** As summarized in Table 3, we conduct ablation studies to assess the contribution of each module in our framework including vision-to-text (V2T), text-to-vision (T2V), and image reconstruction (Img Rec), using both NLG metrics and CE metrics. Also, CLIP-based contrastive learning, a widely adopted approach in the literature, is implemented as a reference.[4] Additionally, the number of epochs required for convergence is reported.

As seen, the baseline model with only V2T achieves moderate performance, capturing basic patterns but lacking clinical detail extraction. Adding CLIP-based contrastive learning (V2T+CLIP) seems to have minimal impact on both IU-Xray and MIMIC-CXR in terms of NLG metrics. It slightly improves CE metrics on MIMIC-CXR, e.g., Bert Score increasing from 0.416 to 0.418, and RadCliQ decreasing from 1.201 to 1.197. These results suggest the complexity of visual representation learning for LLM integration, where a CLIP-based feature alignment is not sufficient. Introducing T2V yields stronger improvements, particularly in clinical efficacy, e.g., on MIMIC-CXR, RadGraph F1

---

[4]Specifically, the visual embeddings extracted by the vision encoder from the input CXR image and the LLM-based word embeddings of the corresponding ground truth report are projected into a 512-dimensional space via their respective linear layers. The CLIP loss is then applied to bring these embeddings close.

Table 4: Evaluation of our T2V module using R2GenGPT's backbone on IU-Xray dataset

| Methods | BLEU-1 | BLEU-2 | BLEU-3 | BLEU-4 | ROUGE | METEOR |
|---|---|---|---|---|---|---|
| R2GenGPT | 0.488 | 0.316 | 0.228 | 0.173 | 0.377 | 0.211 |
| R2GenGPT+T2V | **0.509** | **0.336** | **0.246** | **0.188** | **0.403** | **0.218** |

increasing from 0.188 to 0.196 and RadCliQ decreasing from 1.201 to 1.178, showing the effectiveness of refining visual embeddings by the embedding generated by the LLM upon the ground turth report. If cross-referencing Figure. 4, introducing T2V better reduces the distances between the visual embedding and the LLM's word embedding space than V2T alone. The full model, combining V2T, T2V, and Img Rec, achieves the best results, converging in just 5 epochs on MIMIC-CXR and 2 epochs on IU-Xray. In Figure 4, this setup achieves the shortest distances between the visual embedding and the LLM's embedding space. In sum, T2V plays a key role in moving visual embeddings toward the LLM's operational range, while Img Rec accelerates optimization and further refines the visual representation.

**Backbone.** In addition, to verify the effectiveness of our proposed bidirectional visual representation learning, we embed our T2V module into the backbone of R2GenGPT, which employs Swin-Transformer as the visual encoder rather than the MAE used in our model. The performance is shown in Table 4. As seen, integrating our bidirectional learning strategy can also significantly improve R2GenGPT, with consistent improvements across different metrics. Specifically, BLUE-4 increases from 0.173 to 0.188, ROUGE from 0.377 to 0.403, and METEOR from 0.211 to 0.218. These results reinforce the advantages of our bidirectional visual feature learning when integrating LLM for R2Gen on different backbone models.

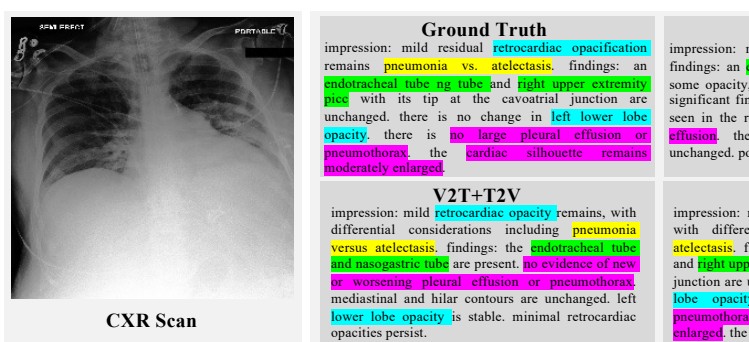

Figure 3: Comparison of the reports generated by the model trained under different settings on MIMIC-CXR. The key medical information are highlighted using different colors (See the text).

**Visualisation.** Figure. 3 visualizes the generated reports for the same CXR image using different model configurations. Key medical terms are highlighted in different colors to compare clinical accuracy and relevance. The report produced by V2T (baseline) provides basic observations but lacks clarity on the differential diagnosis for the mild retrocardiac opacity. With the addition of the T2V module (V2T+T2V), the generated report improves by clearly stating the differential diagnoses of pneumonia versus atelectasis and confirming that there is no evidence of new or worsening pleural effusion or pneumothorax, while also indicating that the left lower lobe opacity is stable. Finally, the full model (V2T+T2V+Rec) generates a more comprehensive and precise summary by reiterating the differential diagnoses and emphasizing the stability of the retrocardiac opacity, as well as detailing the unchanged status of the endotracheal and nasogastric tubes, and the PICC line, aligning closely with the ground-truth report.

To further investigate the characteristics of the learned visual representation, we calculate the minimal Euclidean distance of each visual embedding from the LLM's built-in word embedding space, and visualize the histograms of such distances obtained from the whole test set of MIMIC-CXR. Specifically, for each CXR image, the visual embeddings extracted by the vision encoder and projected by visual mapper are compared with all token embeddings within the LLM's built-in embedding space. The Euclidean distance of each visual embedding to the closest token is recorded and

the distribution of these minimum distances across all 3858 test samples of MIMIC-CXR is visualized in the histogram in Figure. 4(a). As seen, by the proposed bidirectional learning, our V2T+T2V method significantly drags the visual embeddings closer to the LLM built-in word embedding space. This trend becomes more pronounced by further incorporating the image reconstruction component (V2T+T2V+REC). In contrast, introducing the CLIP-like contrastive loss (V2T+CLIP) does not substantially reduce the minimum distances. These results are consistent with the performance observations in Table 3. The similar conclusion can be drawn from the IU-Xray dataset in Figure. 4(b).

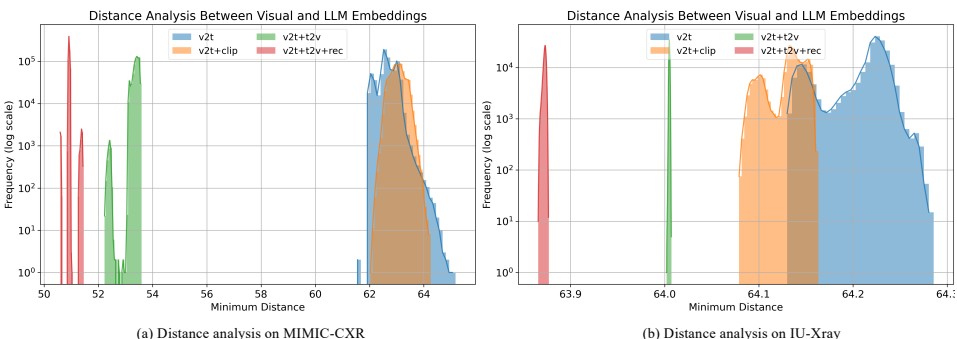

(a) Distance analysis on MIMIC-CXR

(b) Distance analysis on IU-Xray

Figure 4: Histograms of the minimum Euclidean distances of the learned visual embeddings to the LLM's built-in word embedding space when different visual representation learning schemes are used. The results are based on all test samples in MIMIX-CXR and IU-XRay datasets, respectively.

## 5 CONCLUSION, LIMITATION, AND FUTURE WORK

As we explore leveraging the powerful capabilities of LLMs to enhance radiology report generation, a key challenge lies in improving the compatibility between the visual representation produced by the vision encoder and the operational scope of the LLM—essentially making the visual data more easily "understood" by the LLM. This work addresses this challenge by integrating both vision-to-text and text-to-vision tasks, resulting in a bidirectional learning framework for radiology report generation. The framework is intentionally built on a frozen LLM, preserving computational efficiency and performance. As demonstrated through experiments, this bidirectional learning approach improves report quality and strengthens the desired compatibility. In a broader sense, by fostering mutual reinforcement of visual and textual embeddings, our work underscores the potential of LLMs as a foundation for cross-modal generation. This paradigm not only advances radiology report generation but also opens new pathways for bridging modality gaps in future research.

Meanwhile, we also observe in our investigation that while LLMs excel at generating coherent and accurate text, the visual embeddings generated by LLMs are insufficient for synthesizing high-fidelity images (e.g., the input chest X-ray scans). This difficulty arises from the modality gap, as LLMs are inherently designed for text generation, complicating the translation of embeddings between textual and visual domains. Directly generating high-quality images from frozen LLMs remains challenging. We anticipate that by fine-tuning LLMs or utilizing truly multimodal large models, we can extend our work to achieve fully bidirectional generation, producing both high-quality reports and high-fidelity images. This will be an intriguing direction for our future work.

Ethical considerations are essential when developing radiology report generation techniques. Current methods are still not sufficiently reliable for practical medical diagnosis. Even as these techniques mature, issues of fairness, transparency, and explainability must be thoroughly addressed. Furthermore, the development of radiology report generation techniques requires large amounts of medical data, making it crucial to fully respect and protect patient privacy during the collection, curation, and use of benchmark datasets.

Our work builds on publicly available models, techniques, benchmark datasets, software packages, and programming languages. The framework, method, and implementation settings are elaborated in the sections of methodology and experimental study. Upon publication, we will release the source code and model checkpoints, along with detailed instructions to ensure full reproducibility.

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

## A    APPENDIX

You may include other additional sections here.

