# OpenReview forum: "Bidirectional Learning for the Visual Representation in Radiology Report Generation with Frozen LLMs"
_ICLR.cc/2025/Conference — ICLR 2025 Conference Withdrawn Submission_

### Official Review · Reviewer_8CGS · 2024-10-29

**Soundness:** 3
**Presentation:** 2
**Contribution:** 2
**Rating:** 5
**Confidence:** 3

**Summary:**

This paper presents a novel bidirectional learning framework for enhancing radiology report generation (R2Gen) by achieving compatibility between the visual representation and the LLM’s word embedding space. It proposes a representation learning framework that integrates both vision-to-text and text-to-vision learning. By incorporating the representations, the model with a frozen LLM  achieves state-of-the-art results on the IU-Xray and MIMIC-CXR datasets, demonstrating improvements in both language accuracy and clinical efficacy. Ablation studies confirm the effectiveness of the bidirectional learning process in aligning visual representations with the LLM’s word embedding space.

**Strengths:**

1. The paper introduces a novel approach that requires that visual representations aid the LLM in report generation (vision-to-text) and, conversely, be generated by the LLM when provided with the ground truth report (text-to-vision).
2. By combining vision-text consistency loss, text-vision consistency loss, and image reconstruction loss,  the visual representation could capture the core features of the input images.
3. Using a frozen LLM and adding no extra computational burden at inference, the method is efficient while maintaining good performance.

**Weaknesses:**

1. For clinical efficacy, the [CheXbert labeling tool](https://github.com/stanfordmlgroup/CheXbert) is usually applied to convert each report into 14 disease classification labels. More metrics, such as precision, recall, and F1, should be used to measure the quality of the generated reports.
2. Additionally, why aren’t SOTA report generation models on IU-Xray like [XPRONET ](https://link.springer.com/chapter/10.1007/978-3-031-19833-5_33).
3. In Table 4, R2GenGPT+T2V achieves better results than the proposed model. It is not clear if other T2V and Img Rec can work for a stronger baseline.
4. The authors claim that the enhanced visual representation could capture the core characteristics of the underlying distribution of these radiology images. However, the experiments were only conducted on the report generation task. More downstream tasks (like Disease Classification) need to be included to enhance the demonstration.
5. The paper did not mention the results of image reconstruction and the effectiveness of the text encoder.

**Questions:**

Could causal attention lead to causal relationships in the generated visual embeddings? It suggested that providing more details about text-to-vision loss.

---

### Official Review · Reviewer_DcWr · 2024-11-01

**Soundness:** 2
**Presentation:** 3
**Contribution:** 2
**Rating:** 3
**Confidence:** 5

**Summary:**

This paper introduces a novel bidirectional learning framework for radiology report generation that enhances the compatibility between visual representations and large language models (LLMs) without modifying the LLM itself. By incorporating both vision-to-text and text-to-vision tasks alongside image reconstruction, the framework aligns visual embeddings with the LLM’s word embeddings, improving language accuracy and clinical efficacy in generated reports. The approach outperforms existing methods on benchmarks like IU-Xray and MIMIC-CXR, showing significant improvements in both linguistic coherence and clinical relevance through bidirectional learning.

**Strengths:**

The paper presents a creative approach to radiology report generation through its bidirectional learning framework, a fresh strategy that integrates vision-to-text and text-to-vision tasks, and an innovative requirement that the visual representation should be generatable by a frozen large language model (LLM). This approach distinguishes itself from previous methods like MedM2G, which used diffusion models to generate other modality images. By leveraging bidirectional learning, this work enhances the compatibility of visual representations with LLMs, pushing the boundaries of automated radiology report generation without altering the LLM itself. In addition, the authors systematically implement vision-to-text and text-to-vision tasks to align visual representations with the LLM’s word embedding space, backed by an image reconstruction task. And this paper is straightforward to follow.

**Weaknesses:**

1. The paper does not provide a comprehensive comparison with existing methods focused specifically on enhancing visual representations for radiology report generation. Given its emphasis on bidirectional learning to strengthen visual representations while keeping the language decoder fixed, it would be beneficial to compare with other works that also target visual representation enhancement. For example, methods utilizing lesion region features or contrastive learning, would provide a more relevant context for evaluating the framework's performance. Including these comparisons could give a clearer indication of the unique contributions and advantages of the bidirectional learning approach over other targeted visual representation strategies.
[1] Tanida, Tim, et al. "Interactive and explainable region-guided radiology report generation." Proceedings of the IEEE/CVF Conference on Computer Vision and Pattern Recognition. 2023.
[2] Li, Mingjie, et al. "Contrastive learning with counterfactual explanations for radiology report generation." European Conference on Computer Vision. Springer, Cham, 2024.

2. The paper employs L2 loss alone for both visual representation reconstruction and image reconstruction, which is known to produce overly smooth and blurry reconstructions with limited detail. Current generative models often show that L2 loss alone is insufficient for capturing fine details, which are particularly crucial for medical images where precise information affects clinical decisions. This simplistic design may lead to suboptimal visual representations, lacking the nuanced details needed in medical contexts. Incorporating more advanced techniques, such as perceptual loss, adversarial loss, or feature-level alignment, could improve the richness of the reconstructed representations, ensuring they retain critical medical details.

**Questions:**

In Table 1, the results for models like R2Gen are drawn from the original papers, whereas in Table 2, the clinical efficacy metrics for these models are reproduced by the authors. Could you clarify why the reproduced NLG metric values were not included in Table 1 as well?

---

### Official Review · Reviewer_KpPb · 2024-11-03

**Soundness:** 2
**Presentation:** 2
**Contribution:** 3
**Rating:** 5
**Confidence:** 5

**Summary:**

The authors propose a method to improve the alignment of representations from a vision encoder with a frozen large language model (LLM)'s embedding space, for the task of radiology report generation. This is achieved by introducing a text-to-image pathway, forcing the visual representations to be such that the LLM is able to predict them from text, and by also including an image reconstruction loss. The approach is evaluated on the MIMIC-CXR and IU-Xray datasets with a variety of common lexical and clinical metrics, appearing to outperform the reported baselines.

**Strengths:**

- The text is clearly written and easy to follow.
- The paper tackles an important problem in radiology report generation (and vision-language learning, broadly): how to improve the alignment between visual and textual representations such that they can be processed by a shared language model. This is even more critical in the scenario considered here, with the language model kept frozen.
- The prospect of achieving competitive performance on this task (or other visual reasoning tasks) while keeping the LLM frozen is encouraging.
- The proposed approach is relatively simple and is able to reuse existing pretrained components.

**Weaknesses:**

1. **Task inputs and outputs:** The choices of inputs and outputs force the model to hallucinate details that would require additional context to predict correctly. Specifically:
   - One cannot write an accurate CXR report from a lateral view alone, as most radiological findings can only be clearly seen on a frontal scan (posteroanterior or anteroposterior). If the model supports only one image, including lateral views in training and evaluation just introduces noise.
   - Relatedly, each report is written for a radiographical study comprising one or more images (at least one of which must typically be frontal). Therefore, metrics for external baselines may not be comparable if they were computed on a study level vs image level.
   - The model was trained to generate both _Findings_ and _Impression_ sections. However, the _Impression_ generally contains clinical interpretation (e.g. the differential diagnoses seen in Fig. 3) and often recommendations for patient management, neither of which can be inferred from an image.
   - The example in Fig. 3 highlights the issue with lack of access to prior images: all of the generated reports shown there contain multiple glaring hallucinations of temporal comparisons (e.g. "stable", "persists", "unchanged", etc.).

2. **Missing baselines:** The comparisons are missing several SOTA LLM-based baselines, for example:
   - Med-PaLM M: Tu et al. (2024). Towards Generalist Biomedical AI. NEJM AI 1:AIoa2300138.
   - LLaVA-Rad: Chaves et al. (2024). Towards a clinically accessible radiology foundation model: open-access and lightweight,
with automated evaluation. arXiv:2403.08002.
   - MedVersa: Zhou et al. (2024). A Generalist Learner for Multifaceted Medical Image Interpretation. arXiv:2405.07988.
   - MAIRA-1: Hyland et al. (2023). MAIRA-1: A specialised large multimodal model for radiology report generation. arXiv:2311.13668.
   - MAIRA-2: Bannur et al. (2024). MAIRA-2: Grounded Radiology Report Generation. arXiv:2406.04449.

   I understand some of these are recent and not formally published yet, but their results seem much better than those reported in this paper on most metrics.

3. **Metrics:** I have various questions and concerns about the evaluation metrics: unclear why clinical metrics were not computed/-able for IU-Xray; ablations were evaluated on IU-Xray only and without clinical metrics; the common CheXpert/CheXbert F1 metric was not included. Overall, there is too much emphasis on the NLG metrics, which are very style-dependent and vastly less important than clinical factuality in practice. In fact, optimising for NLG metrics without the relevant inputs will force the model to memorise spurious correlations.

4. **Incomplete ablations:** Missing a setting with only V2T + Rec (without T2V), as it's not clear how much of the improvement seen in Table 3 and Fig. 4 is due to T2V or Rec in isolation. This is an important baseline, as T2V requires an extra model to train, loss term, and forward pass on the LLM. Also, because the goal is to align the embedding spaces, I believe the "CLIP-like" baseline would be stronger and more informative with an explicit Euclidean distance objective, rather than contrastive.

**Questions:**

1. Please comment on the limitations and implications of the current task inputs/outputs. This discussion must also be included in the paper.
2. Regarding the missing external baselines, the authors should consider either: (i) including results for at least some of them and discussing the key differences and relative limitations of the proposed approach; or at least (ii) citing them and explaining why a direct comparison was not performed.
3. I strongly suggest to consistently include results on both MIMIC-CXR and IU-Xray across both NLG and clinical metrics for all experiments.
4. Consider computing additional clinical efficacy metrics like one of the variants of CheXpert/CheXbert F1 to improve evaluation.
5. Consider running the suggested ablation experiments, if you can, as I believe they could significantly strengthen the paper. Other optional but possibly informative ablations include: (i) freezing also the vision encoder to evaluate the effect of aligning only the mapper, and lastly (ii) unfreezing the LLM as a performance upper bound.
6. Did the authors train a single model on a mixture of MIMIC-CXR and IU-Xray? If so: in what proportion? was it trained sequentially? how were the batches mixed? If otherwise there were separate models for each dataset, this really must be clarified.

---

### Official Review · Reviewer_1Qbj · 2024-11-03

**Soundness:** 2
**Presentation:** 3
**Contribution:** 2
**Rating:** 3
**Confidence:** 4

**Summary:**

This paper introduces a bidirectional learning, both vision-to-text and text-to-vision, framework for radiology report generation task. The framework using frozen large language model integrating vision-to-text, text-to-vision, and image reconstruction components to improve the visual representation of LLM embedding space. And it finally improve the linguistic accuracy and clinical relevance of generated report.

**Strengths:**

The paper seeks to integrate multiple strategies within its framework, such as vision-to-text and text-to-vision transformations. This bidirectional learning approach is a promising endeavor aimed at enhancing visual representation learning by leveraging the strengths of both modalities. The authors further emphasize computational efficiency by employing a frozen large language model (LLM), a practical decision that aligns with contemporary trends in optimizing model performance. This choice not only reduces computational overhead but also increases the framework’s applicability in real-world scenarios. Additionally, the authors conducted a comprehensive series of evaluation experiments on widely recognized benchmark datasets, IU-Xray and MIMIC-CXR, offering a thorough assessment across various metrics. This extensive evaluation underscores the robustness and practical relevance of their proposed framework.

**Weaknesses:**

The proposed framework seems to function more as a combination of various components rather than a fully integrated system. By designating certain modules as either trainable or frozen, the entire process appears somewhat fragmented. This fragmentation is particularly evident in the image reconstruction module, whose integration feels rather superficial and not deeply embedded into the overall training pipeline. Despite these concerns, the ablation studies indicate that components like the text-to-vision (T2V) module and the image reconstruction module only yield marginal improvements to the final results. This suggests that while these components contribute to the model's functionality, their impact on performance may not be as substantial as initially expected, raising questions about their necessity and effectiveness within the broader framework.

**Questions:**

1. Why not use generated visual embeddings to reconstruct the image? It seems like in this way you can integrate the reconstruction loss into the whole training process.

2. Have you tested the V2T with image reconstruction in the ablation study? Because you use the extracted visual embedding for the reconstruction, you can test the performance in the situation only using REC loss and VTC loss.

---

### Official Review · Reviewer_ACoS · 2024-11-03

**Soundness:** 2
**Presentation:** 3
**Contribution:** 2
**Rating:** 5
**Confidence:** 3

**Summary:**

This paper introduces a novel learning method for radiology report generation using a bidirectional framework. The framework aims to enhance visual representation learning by integrating the image-to-text generation task with regularizations, including a text-to-visual embedding task and an image reconstruction task. The approach incorporates a learnable visual encoder, a visual decoder, and a text encoder, while keeping an LLM frozen. The method demonstrates state-of-the-art performance on the IU-Xray and MIMIC-CXR datasets in the authors' manuscript, showing improvements in both language accuracy and clinical efficacy.

**Strengths:**

- The manuscript targets a critical part of radiology report generation by utilizing large language models (LLMs), emphasizing the importance of aligning the LLM with image embeddings.
- The use of a frozen LLM to generate visual embeddings for regularizing training is novel.
- The manuscript is well-written in fluent language.

**Weaknesses:**

- The motivation and necessity for generating visual embeddings through a frozen LLM are not well justified.
- The results lack several classical and reliable clinical efficacy (CE) metrics, such as CheXbert precision, recall, and F1, which are reported in most methods.
- Certain implementation details need further clarification.

Please refer to the questions section for more detailed concerns.

**Questions:**

**Major**：

1. The main motivation and necessity for generating visual embeddings through the LLM are not obvious and should be justified. Many models, especially those designed for discriminative tasks, do not require the ability to generate inputs to perform well. (Humans are also good at detecting objects from photos although they can not draw them well)
2. Why keep the LLM frozen? For a pretrained LLM like LLaMA 2, generating visual embeddings seems to be an out-of-distribution task. Is it appropriate to rely solely on a learnable text encoder to learn this task?
3. Since the LLM is frozen, are the extracted text embeddings and generated visual embeddings comparable or explainable through the LLM? For example, can those embeddings be projected back to tokens using the LM head? Or, are these embeddings close to the LLM's original vocabulary embeddings?
4. In Figure 2(b), why are visual embeddings added in the text-to-vision branch? If the goal is to generate visual embeddings, why include these embeddings in the input?
5. Implementation Details: The manuscript states that images are randomly cropped to 224 x 224 pixels. The original resolution of MIMIC-CXR images can be around 3000 x 2500 pixels. Does this mean that only a small patch of the image is being cropped?
6. In the results section, an important clinical efficacy metric, the CheXpert metrics, is missing. Many comparable methods, such as R2GCMN, CvT2DistilGPT2, and R2GenGPT in the manuscript, reported their CheXpert classification precision, recall, and F1 scores for certain diseases, or the macro/micro averages for 5 or 14 diseases. This is one of the most reliable, direct, and explainable metrics for clinical efficacy. Please explain why this metric was omitted.
7. Table 3: In the ablation study, the V2T-only setting achieves results on IU-Xray that surpass all other methods in BLEU-4, ROUGE, and METEOR, and ranks first in METEOR, second in BLEU-4, RadGraph-F1, BERTScore, and RadCliQ, and third in ROUGE. The V2T-only setting appears to be a straightforward vision-language model with a learnable vision encoder and a frozen LLM, yet it outperforms or matches most compared state-of-the-art results. How should this result be interpreted?
8. Visualization section and Figure 3: The sentence contains phrases like "no evidence of new or worsening," "stable," and "unchanged," which are presented as examples of a more comprehensive and precise summary in optimized methods. However, these terms describe the dynamic state of a lesion or condition. Given that the input is a single frame of an image, are these descriptions more likely to be hallucinations rather than accurate reporting?

**Minor**:

1. Line 190 mentions Figure 3(a), which does not appear in the manuscript. Is this a typo?

---

### Note · Authors · 2024-12-03

I have read and agree with the venue's withdrawal policy on behalf of myself and my co-authors.